# Pharmacokinetic profile of injectable tramadol in the koala (*Phascolarctos cinereus*) and prediction of its analgesic efficacy

**Benjamin Kimble**[1©], **Larry Vogelnest**[2©], **Peter Valtchev**[3©], **Merran Govendir**[1©]*

**1** Sydney School of Veterinary Science, The University of Sydney, Sydney, New South Wales, Australia, **2** Taronga Conservation Society Australia, Mosman, New South Wales, Sydney, New South Wales, Australia, **3** School of Chemical and Biomolecular Engineering, The University of Sydney, Sydney, New South Wales, Australia

© These authors contributed equally to this work.
* merran.govendir@sydney.edu.au

**Data Availability Statement:** All relevant data are within the paper and its Supporting information files.

## Abstract

Tramadol is used as an analgesic in humans and some animal species. When tramadol is administered to most species it undergoes metabolism to its main metabolites M1 or O-desmethyltramadol, and M2 or N-desmethyltramadol, and many other metabolites. This study describes the pharmacokinetic profile of tramadol when a single subcutaneous bolus of 2 mg/kg was initially administered to two koalas. Based on the results of these two koalas, subsequently 4 mg/kg as a single subcutaneous injection, was administered to an additional four koalas. M1 is recognised as an active metabolite and has greater analgesic activity than tramadol, while M2 is considered inactive. A liquid chromatography assay to quantify tramadol, M1 and M2 in koala plasma was developed and validated. Liquid chromatography-mass spectrometry confirmed that M1 had been identified. Additionally, the metabolite didesmethyltramadol was identified in chromatograms of two of the male koalas. When 4 mg/kg tramadol was administered, the median half-life of tramadol and M1 were 2.89 h and 24.69 h, respectively. The M1 plasma concentration remained well above the minimally effective M1 plasma concentration in humans (approximately 36 ng/mL) over 12 hours. The M1 plasma concentration, when tramadol was administered at 2 mg/kg, did not exceed 36 ng/mL at any time-point. When tramadol was administered at 2 mg/kg and 4 mg/kg the area under the curve M1: tramadol ratios were 0.33 and 0.50, respectively. Tramadol and M1 binding to plasma protein were determined using thawed, frozen koala plasma and the mean binding was 20% and 75%, respectively. It is concluded that when tramadol is administered at 4 mg/kg as a subcutaneous injection to the koala, it is predicted to have some analgesic activity.

## Introduction

Wild koalas may be injured by car strikes, bushfires or animal attacks and are susceptible to infectious diseases such as Chlamydiosis, all potentially requiring analgesia to improve the

**Funding:** BK was supported by The Koala Hospital, Port Macquarie, NSW, Australia and the Winifred Violet Scott Charitable Foundation. The funders had no role in study design, data collection and analysis, decision to publish, or preparation of the manuscript.

**Competing interests:** The authors have declared that no competing interests exist.

quality of patient recuperation and survival. The only published pharmacokinetic (PK) profile for any analgesic in the koala has been for the non-steroidal anti-inflammatory drug (NSAID) meloxicam which has a short elimination half-life of 1.19 hour (hr) (range 0.71 to 1.62 hr) [1, 2], compared with 24 hr in dogs [3] and approximately 13 hr in humans [4]. Therefore, in contrast to many other species, meloxicam would require frequent dosing during a 24 h period for koalas which is problematic for a wild, non-domesticated species. There is therefore a need to identify analgesics that are not only efficacious, but also have a longer duration of analgesia to minimise animal handling.

Tramadol is a centrally acting analgesic used in some species to control moderate to moderately severe pain [5]. In Australia, tramadol is scheduled by the therapeutic goods regulator as a *Prescription Only Medicine*. Its mechanism of action is mediated through interactions with opioid, adrenergic, and serotonin receptors [6, 7]. A significant portion of tramadol's analgesic properties are reported to be derived from the actions of its main metabolite O-desmethyltramadol (O-DSMT, also referred to as 'metabolite 1' [M1]). M1 is reported to have 300 times greater affinity for the mu (μ) opioid receptors than tramadol [8, 9]. Many other metabolites have also been identified, the number of which, varies with species [8, 10]. Another frequently recognized metabolite is N-desmethyltramadol [M2] which is considered inactive [8, 10]. An illustration summarising tramadol's and its major metabolites' metabolism in the human hepatocyte is found at https://www.pharmgkb.org/pathway/PA165946349 [11]

Tramadol is likely to be used more frequently as an analgesic for cats, than dogs, due to a greater conversion of tramadol to M1 (M1: tramadol area under the curve [AUC] ratio is 1–1.2 in the cat) [5]. There is a lower ratio in humans—0.25 [12] and in dogs—0.003 [10]. Due to the necessity to find analgesics that are efficacious for koalas and that tramadol has been used anecdotally in this species [13], the aim of this study was to describe the PK profile of tramadol when injected as a single subcutaneous injection and to predict likely efficacy.

## Materials and methods

### Animals

Six, mature, clinically normal koalas (3 males, 3 females) with body weights from 6.8 to 9.0 kg, (median 7.58 kg), were recruited from the Taronga Zoo colony (Mosman, NSW, Australia). These koalas were considered clinically normal based on regular physical examinations, and haematology and biochemical analyte values. During the study, koalas were housed singly and supplied with various *Eucalyptus* spp. foliage and water *ad-libitum*. The University of Sydney Animal Ethics committee approved this protocol (2015/877) as well as the Taronga Conservation Society of Australia, Animal Ethics Committee (protocol 3a/10/18).

### Drug administration and blood collection

The koalas were anaesthetised with alfaxalone (Alfaxan, Jurox, Pty Ltd, Rutherford, NSW, Australia) at 3 mg/kg administered intramuscularly (i.m.) and maintained under anaesthesia on isoflurane in 100% oxygen via a face mask for placement of a 20 gauge, 1 ¼ inch intravenous catheter into the cephalic vein. A short T-connector extension set (Codan, Santa Ana, CA, USA) and cap was attached to the catheter and flushed with heparinised saline. The extension tube, cap and catheter were secured with tape and bandage, for serial blood collection [14]. Blood (5 mL) was collected at the time of anaesthesia in a lithium heparin tube to establish baseline haematology and biochemistry (designated t = 0 hr). Tramadol hydrochloride (Tramadol solution for injection, Sandoz Pty Ltd, Macquarie Park, NSW) was initially administered to two koalas (one of each sex) at a dose of 2.0 mg/kg as a single s.c. bolus. The 2.0 mg/kg injectable dose was initially selected because it is a suggested dose used in feline veterinary

practice [15]. As many drugs such as chloramphenicol [16] enrofloxacin [17, 18] and meloxicam [1] have superior s.c. absorption compared to the oral route, the s.c. route was selected for this study. On review of the resulting tramadol and M1 plasma concentrations from 2 mg/kg tramadol administration, the Animal Ethics committees approved 4.0 mg/kg s.c. administration to a further four mature koalas (two of each sex).

To determine tramadol, M1 and M2 plasma concentrations, serial blood samples (up to 2 mL) were collected into lithium heparin tubes at t = 0.25, 0.5, 1, 2, 4, 6, 8 and 12 hr after drug administration. The cap, extension tube and catheter were flushed with heparinised saline before and after each collection. The first 0.5 mL of blood was discarded to avoid dilution of samples with heparinised saline. Samples were centrifuged within 1 hr of collection; the plasma was immediately stored at –80˚C and protected from light until drug quantification.

### Drug analysis method and sample processing

**Chemicals.** Tramadol hydrochloride, M1, phosphoric acid sodium hydroxide tert-butyl methyl ether and triethylamine were purchased from Sigma-Aldrich (Castle Hill, NSW, Australia). M2 was purchased from MyBioSource Inc. (San Diego, CA, USA). High pressure liquid chromatography (HPLC) grade methanol, acetonitrile and ethyl acetate were purchased from Thermo Fisher Scientific (Macquarie Park, NSW, Australia).

**Drug analysis.** Quantification of tramadol, M1 and M2 concentrations in plasma by HPLC with fluorescence detection was modified from a previously published method [19]. The components of system included a Shimadzu LC-20AT solvent delivery unit, DGU-20A degasser, a SIL-20A auto injector and a RF-10AXL fluorescence detector (Shimadzu, Rydalmere, NSW, Australia). Shimadzu LC solutions software (Kyoto, Japan) was used for chromatographic control, data collection and data processing. Chromatographic separation was performed with a Synergi 4 μm MAX-RP 80 A, 150 × 4.6mm (Phenomenex, Lane Cove, NSW, Australia) with a 1-mm Opti-guard C-18 pre-column (Choice Analytical, Thornleigh, NSW, Australia) at ambient temperature. The isocratic mobile phase consisted of 0.01 M phosphate buffer and acetonitrile (82.5:17.5, v/v), containing 0.1% triethylamine adjusted to pH 3, at a flow rate of 0.8 mL/min. Fluorescence detection excitation and emission wavelengths were 200 nm and 301 nm, respectively. Standard curve concentrations, ranging from 15.62 to 500 ng/mL, and quality control samples (QCs) (at concentrations of 15.63, 62.5 and 500 ng/mL) were prepared in blank pooled koala plasma collected from a minimum of three koalas. For sample preparation 200 μL of each of the following: plasma samples collected from koalas administered tramadol, standards prepared for the standard curves and the QC samples were mixed with 50 μL of 1 M sodium hydroxide and extracted twice with 800 μL of ethyl acetate / tert-butyl methyl ether (1:1, v/v). The extracted organic portion was dried under vacuum in a Speed Vac Concentrator (Thermo Scientific, MA, USA) at 30 ˚C for 1 h, reconstituted with 200 μL of mobile phase, and finally, 10 μL of the reconstituted sample was injected into the HPLC system. External standard curves were prepared individually for tramadol, M1 and M2 (all with $r^2 > 0.99$) and a weighting factor ($1/x$) to ensure that the relative importance of observations in the regression, especially the larger ones, were not over-fitted. After an extensive search, a suitable internal standard could not identified due to the tramadol, or metabolites' peaks or plasma endogenous peaks [20] obscuring all potential internal standards. Therefore, an external standard was used that comprised of a known concentration of tramadol, M1 and M2 added to untreated pooled koala plasma. Based on the lowest concentration where the precision and accuracy were <15% and within 20% of nominal concentration, respectively, the lower limit of quantification (LLOQ) was calculated as 15.63 ng/mL for tramadol and its

metabolites [21]. The accuracy, precision, and average drug recovery of the three QC samples, all performed in triplicate, are provided in S1 Table in S1 File.

**LC-MS identification of M1.** Although M1 had the same retention time as the analytical standard by liquid chromatography (LC), the structure of the M1 metabolite was confirmed with liquid chromatography—mass-spectrometry (LC–MS) with a system that consisted of a Shimadzu LC–MS 2010EV module (Shimadzu, Kyoto, Japan) and a Phenomenex Gemini $C_{18}$ 5 μm (150 mm × 2 mm) column (Phenomenex, Lane Cove, NSW, Australia) attached to a 1 mm Opti-guard C-18 pre-column. The mobile phase consisted of water and acetonitrile (95: 5, v/v) with 0.1% formic acid. The flow rate was 0.4 mL/min. Detection was accomplished in electrospray ionization (ESI) ion source operated in positive ion mode with scanning range of 150–500 m/z in scan mode, with scan speed of 1000 amu/s, interface voltage mode of 2 kV, interface temperature at 200 ˚C, and flow rate of nitrogen as a nebulizing gas of 1.2 L/min. Retention time of protonated M1 standard was 1.45 min with a m/z of 250.0 as would be expected for M1. LC-MS was also used to analyse the structure of an additional metabolite at a retention time of 3.20 minutes with a molecular weight (MW) of 235.0 and was identified as didesmethyltramadol.

## Pharmacokinetic analysis

Tramadol, M1 and M2 concentration versus time data underwent PK non-compartmental analysis. The peak concentration ($C_{max}$) and the time this was reached ($T_{max}$) was obtained directly from the measured concentrations. The terminal half-life ($t_{1/2}$) was determined by ln 2/$k$ where k is the elimination rate constant which is the inverse slope of the elimination or terminal part of the semi-log curve. The area under the concentration-time curve ($AUC_{0-t\ last}$) was calculated to the last measurable concentration ($t_{last}$) using the log-linear trapezoidal method. The AUC and AUMC from the last observed concentration to infinity were determined by

$$AUC_{t-\infty} = C_{last}/k_{el}$$

$$AUMC_{t-\infty} = (C_{last} \times t_{last}/k_{el}) + (C_{last}/k_{el}^2)$$

The mean residence time (MRT), apparent clearance (Cl/F), apparent volume of distribution (V/F) were determined by the following equations:

$$MRT = AUMC_{0-\infty}/AUC_{0-\infty}$$

$$Cl/F = Dose/AUC_{0-\infty}$$

$$V/F = CL/k_{el}$$

F = bioavailability of the subcutaneous route, remained undetermined as there was no intravascular administration. PK Solver [22] was used to calculate some PK indices for the noncompartmental analysis.

## Tramadol and metabolites' binding to plasma proteins

The percentage of tramadol (at concentrations of 150 and 500 ng/mL) and its main metabolites (100 and 50 ng/mL for M1 and M2, respectively) bound to koala plasma proteins were determined using the ultrafiltration method [23] using a modified protocol [1]. Mixtures of tramadol and metabolites (500 ng/mL of tramadol and 100 ng/mL of both metabolites; or 150 ng/

mL of tramadol and 50 ng/mL of both metabolites) were added to 1 mL of pooled blank koala plasma (n = 6), and adjusted to pH 7.4, were incubated in a water bath at 37 ˚C for 30 min. Then 200 μL of plasma was removed for determination of the total concentrations (Drug $_{total}$) and the remaining plasma was transferred to the reservoir of the Centrifree ultrafiltration device (Merk Millipore, Macquarie Park, Australia) with a membrane MW cut-off of 30 kDa. The ultrafiltration device was centrifuged at 1,500 x $g$ for 10 min at 37 ˚C. After centrifugation, the filtrate was used to determine the free concentrations (Drug $_{free}$). Both fractions were analysed by HPLC as described above. All samples were analysed as duplicates. The percentage of substrate binding to plasma proteins was estimated as 100 - [(Drug $_{free}$/Drug $_{total}$) × 100]. The same concentrations of tramadol and metabolites were added to phosphate buffer saline and underwent ultrafiltration to determine non-specific binding for tramadol, M1 and M2 to the filtration membrane. The non-specific binding was < 0.5% for all substrates.

### Statistical analysis

In order to assess any difference in $C_{max}$ or $AUC_{0-t\ h}$ when normalised for dose (2 mg/kg or 4 mg/kg tramadol administration); the $C_{max}$ / dose and $AUC_{0-t}$ / dose values for tramadol, M1 and M2 were compared by a one-way analysis of variance (ANOVA) using Graphpad Prism 9.0 (San Diego, CA). The level of significance (p) was < 0.05.

## Results

Tramadol, M1 and M2 were detected at approximately 7.69, 3.17 and 8.19 min, respectively, as illustrated in Fig 1. Additionally, many unidentified metabolites were observed in those chromatograms (Fig 1C and 1D). An additional peak with a retention time of 3.20 min was detected in chromatograms in two of the male koalas only (Fig 1C). Using LC-MS, total ion mass (positive ion mode) of this peak was determined as didesmethyltramadol.

The plasma concentrations of tramadol at each time point are provided in S2 Table in S1 File and presented in Fig 2. The plasma concentrations of tramadol and M1 plotted as semi-logarithmic (concentration plotted as log 10) graphs are presented as Figs 3 and 4, respectively. M1 and M2 could both be detected at the first time point i.e. 15 minutes after tramadol administration. M1 and M2 concentrations at each time point are provided in S3 and S4 Tables in S1 File, respectively. M1 and M2 had median maximal plasma concentrations ($C_{max}$) of 88.82 (illustrated in Fig 4) and 132.76 ng/mL (due to reported inactivity of M2, it is not represented as a Fig). A summary of the PK indices of tramadol, M1 and M2, as determined by non-compartmental analysis, are presented in Tables 1–3, respectively. Binding to koala plasma proteins (average ± SD %), tramadol, M1, and M2 were 16.00 ± 3.98%, 75.20 ± 0.29%, and 31.29 ± 9.47%, respectively. For both dosages, there were no reported side effects other than the animals were slightly sedated over the first two to six hours, but resumed their normal demeanour, thereafter.

There were no significant differences in the tramadol, M1 or M2 $C_{max}$ or $AUC_{0-t}$ when normalised for tramadol dose (2 mg/kg or 4 mg/kg administration).

## Discussion

Tramadol has a complex PK profile which appears to be species specific. Tramadol generates many metabolites, the number of which varies between species [10]; its quantifiable metabolites each exist as enantiomers; and each enantiomer may have a different rate of elimination [19, 25]; and both fast and slow tramadol metabolisers may exist within a species [26]. Despite its inherent pharmacological complexity, tramadol is administered as an analgesic for koalas

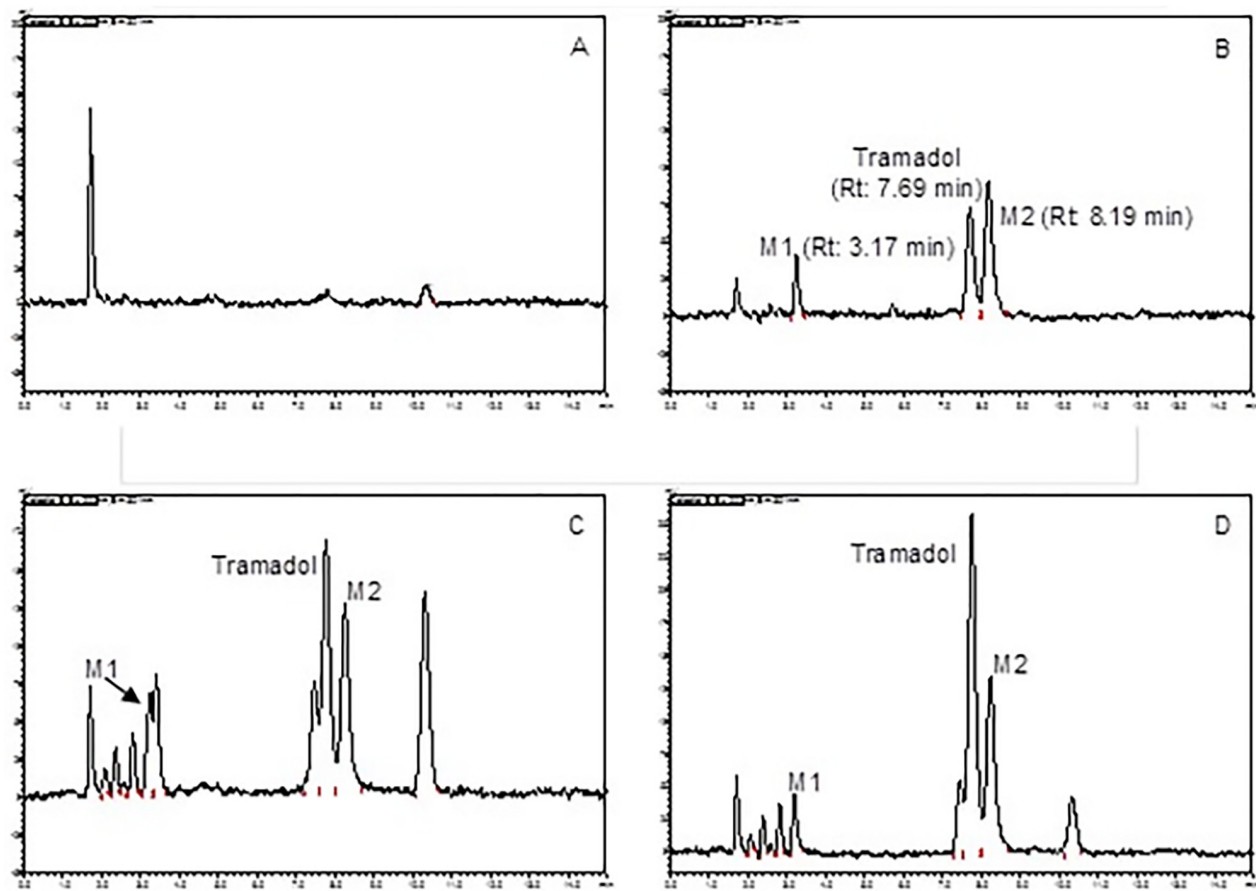

**Fig 1. Chromatograms of tramadol and its metabolites in koala plasma.** A) blank koala plasma; B) blank plasma spiked with tramadol, M1, and M2 (all at a concentration of 62.5 ng/mL); C) male koala plasma 2h after 4 mg/kg tramadol administration; D) female koala plasma 2 h after 4 mg/kg tramadol administration. Additional unidentified metabolites can be seen in C and D.

[13], and this is the first study to describe some aspects of tramadol's PK profile in this species and provides valuable observations.

Initially, the more conservative dose of 2 mg/kg was administered, however at that dosage, the median maximal M1 plasma concentration (37.31 μg/mL) was similar to the minimal effective M1 plasma concentration in humans of approximately 36 ng/mL [10, 27, 28], as illustrated in Fig 4. Consequently, the 4 mg/kg s.c. dosage was administered to four koalas which resulted in the M1 plasma concentration exceeding 36 ng/mL for over 12 hours. One of the limitations of interpretation of the M1 data was that the last data point collected was at 12 h, indicating that at 12 h, M1 was in the early elimination phase in all koalas. Further plasma collection at time points beyond 12 h of tramadol administration would provide a more accurate determination of the M1 plasma profile. On the basis of the available data, M1 had a longer median half-life (2 mg/kg: 14.07 h; 4 mg/kg: 24.69 h) than tramadol (2 mg/kg: 2.87, 4 mg/kg: 2.89 h). M1's half-life exceeding that of its parent is reported for other species: the human M1 half-life = 6 to 7 h, and tramadol half-life = 5 to 6 h [12]; the feline M1 half-life = approximately 4 h, and tramadol half-life = 2.5 h [5]; and the canine M1 half-life = 1.7 h, and tramadol half-life = 0.8 h [10, 29]. There were no statistical differences when the $C_{max}$ or $AUC_{0-t}$ for tramadol, M1 or M2 were normalised for tramadol dose, suggesting that tramadol, M1 and M2 demonstrate linear pharmacokinetics regardless of tramadol dose.

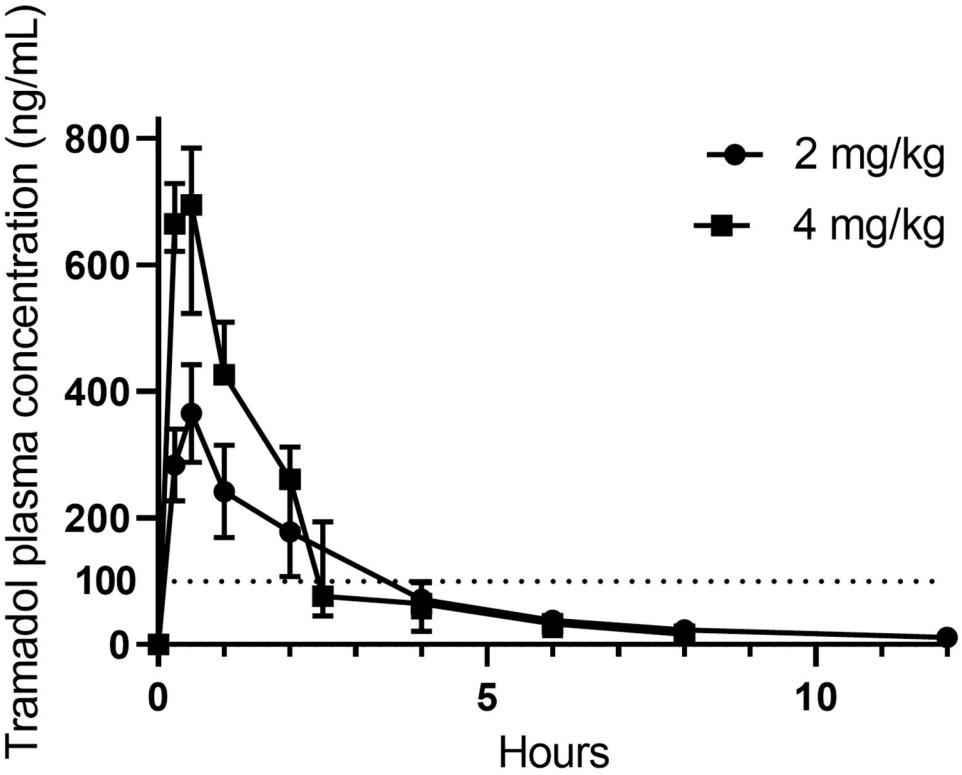

**Fig 2. Median (with error) tramadol plasma concentrations vs. time curves when administered at 2 mg/kg, s.c. (n = 2) and 4 mg/kg, s.c. (n = 4).** Dotted line signifies suggested tramadol minimally effective plasma concentration in humans of 100 ng/mL [24]. The graph of the 4 mg/kg tramadol dose dips slightly below the 2 mg/kg tramadol dose–this may be due to individual variability between koalas.

Tramadol, M1 and M2 binding to plasma protein was undertaken with thawed / frozen koala plasma which has limitations [30, 31]. Generally, tramadol's binding to human plasma proteins is approximately 20% [12] and the bound fraction of M1 to human plasma proteins can be as high as 80% [25], similar to that observed with koala plasma proteins. In contrast, the mean percentage of tramadol and M1 binding to feline plasma proteins is reported as 15% and 17%, respectively [32].

Based on $T_{max}$, s.c. tramadol absorption was rapid and comparable between koalas (median $T_{max}$ = 0.38) and humans (0.34 ± 0.31 h after 50 mg s.c. bolus) [33]. The mean [median] apparent volume of distribution of tramadol in koalas (8.60 ± 2.33 [7.70] L/kg) was of the same order of magnitude, but greater than that reported in other species (Table 4). Tramadol is reported to have high tissue affinity [10] and tramadol may have a high tissue affinity in the koala.

The mean [median] apparent clearance of tramadol itself, in koalas was 39.40 ± 13.36 [38.04] mL/min/kg, comparable to that in cats (20.8 ± 3.2) [5], dogs (54.63 ± 8.19) [10], and horses (26 ± 3) [34], but faster than in humans (7 to 8 mL/min/kg) [33]. A summary of tramadol parameters and M1 half-life available for some species are provided in Table 4.

In humans, tramadol is transformed by the highly polymorphic enzyme cytochrome P450 CYP2D, and to a lesser extent by CYP3A4, to M1 [11, 35]. Furthermore, in humans CYP2D is also involved in the further metabolism of M1 to M5, and M2 to M3 [11]. The activity of these enzymes, particularly CYP2D, can vary among individuals within a species [35, 36]. In

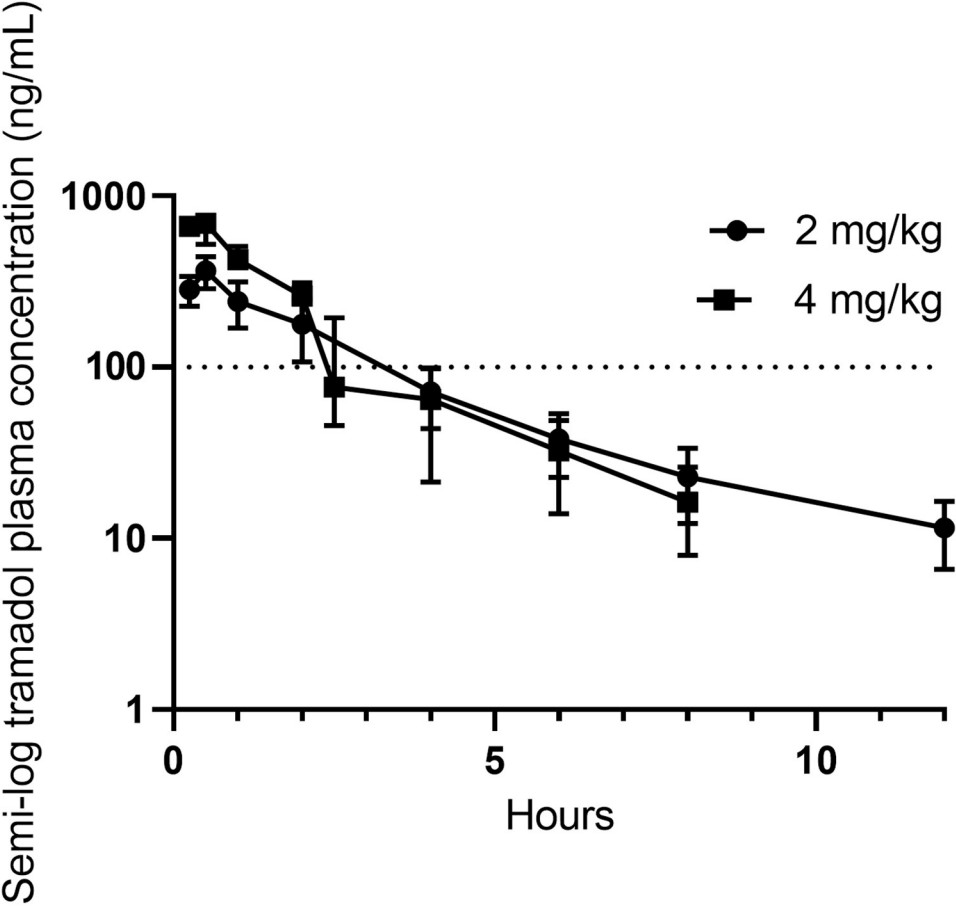

**Fig 3. Median (with error) semi-log tramadol plasma concentration versus time when administered at 2 mg/kg, s. c. (n = 2) and 4 mg/kg, s.c. (n = 4).** Dotted line signifies tramadol minimally effective plasma concentration of 100 ng/ mL [24]. The graph of the 4 mg/kg tramadol dose dips slightly below the 2 mg/kg tramadol dose–this may be due to individual variability between koalas. The tramadol concentrations of the 4 mg/kg tramadol dose at not provided at 12 h as they are below the LLOQ = 15.63 ng/mL.

humans, some individuals are 'ultra-metabolisers', 'extensive' or 'poor' tramadol metabolisers who produce high, intermediate or lower amounts of M1, respectively [36]. Due to the low numbers of subjects in this study, it is not possible to state whether there are ultra-, extensive, or poor tramadol metabolisers in koalas. However, koala 'K3' had a much faster elimination of tramadol, M1 and M2 compared to the others. K3 may have had very active metabolism by enzymes with 'CYP2D6-like' activity compared to the others.

Different tramadol metabolism rates have been associated with sex. In a recent study involving tramadol administration in dogs undergoing routine neutering the $C_{max}$ was greater and $T_{max}$ later, in male dogs [37]. In the chromatograms of two of the male koalas it appeared that M1 had two peaks (Fig 1C). This peak pattern was not seen in those of the other male koala (K4) or the females. The second peaks associated with M1 were identified by LC-MS as didesmethyltramadol. Further investigation with more koalas, of both sexes, is required to confirm that generation of this metabolite is associated with the animal's sex.

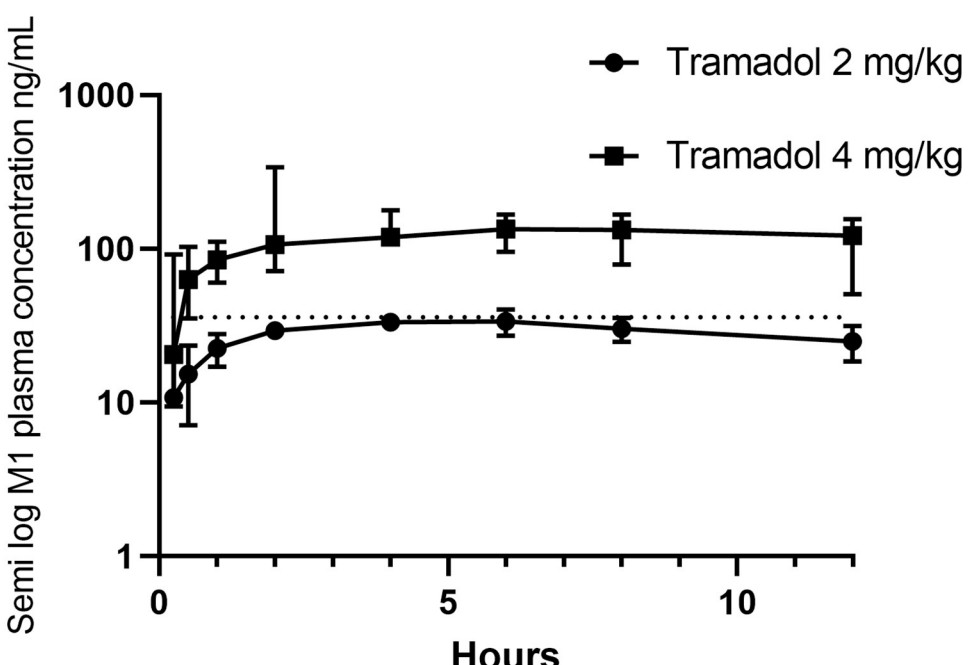

**Fig 4. Median (with error) semi-log M1 plasma concentrations vs. time curves when administered at 2 mg/kg, s.c. (n = 2) and 4 mg/kg, s.c. (n = 4).** Dotted line signifies suggested M1 minimally effective plasma concentration in humans of 36 ng/mL [24].

**Table 1. Pharmacokinetic indices of tramadol when administered at 2 mg/kg, and 4 mg/kg, both administered by s.c. injection.**

| | Tramadol administered @ 2 mg/kg | | | Tramadol administered @ 4 mg/kg | | | | | Combined 2 & 4 mg/kg* |
|---|---|---|---|---|---|---|---|---|---|
| Indice | Koala 1 (K1) (M) | K2 (F) | Median | K3 (M) | K4 (M) | K5 (F) | K6 (F) | Median | Median |
| **Koala body weight (kg)** | 9.0 | 6.85 | 7.93 | 7.90 | 7.65 | 6.95 | 7.50 | 7.58 | 7.58 |
| **Tramadol dose (mg)** | 18.2 | 13.70 | 15.95 | 31.60 | 30.60 | 27.80 | 30.0 | 30.3 | |
| $k_{el}$ **(1/h)** | 0.32 | 0.19 | 0.26 | 0.53 | 0.20 | 0.20 | 0.25 | 0.23 | 0.23 |
| $t_{1/2}$ **(h)** | 2.16 | 3.58 | 2.87 | 1.31 | 3.53 | 3.39 | 2.39 | 2.89 | 2.89 |
| $T_{max}$ **(h)** | 0.25 | 0.5 | 0.375 | 0.25 | 0.5 | 0.5 | 0.25 | 0.38 | 0.38 |
| $C_{max}$ **(ng/mL)** | 387.81 | 503.74 | 445.78 | 707.85 | 843.13 | 784.89 | 728.85 | 756.87 | |
| $C_{max}$ **/ dose (ng/mL)/mg** | 21.31 | 36.77 | 27.95 | 22.40 | 27.55 | 28.23 | 24.30 | 24.98 | p = 0.54[a] |
| $AUC_{0-12h}$ **(ng/mL*h)** | 692.37 | 1407.24 | 1049.81 | 1066.38 | 1831.34 | 1507.24 | 2003.83 | 1669.29 | |
| $AUC_{0-inf}$ **(ng/mL*h)** | 735.72 | 1504.19 | 1119.96 | 1092.63 | 1958.58 | 1586.52 | 2093.69 | 1772.55 | |
| $AUC_{0-inf}$ **/ dose (ng/mL*h)/mg** | 38.04 | 102.72 | 65.82 | 33.75 | 58.85 | 54.22 | 66.79 | 55.09 | p = 0.50[a] |
| $AUMC_{0-inf}$ **(ng/mL*h^2)** | 1929.46 | 5968.69 | 3949.08 | 2029.25 | 7412.53 | 5229.76 | 7491.21 | 6321.15 | |
| **MRT (h)** | 2.62 | 3.97 | 3.30 | 1.86 | 3.78 | 3.30 | 3.58 | 3.44 | 3.44 |
| **Vz/F (L/kg)** | 8.49 | 6.86 | 7.68 | 6.90 | 10.41 | 12.32 | 6.58 | 8.66 | 7.70 |
| **Cl/F (L/kg/h)** | 2.72 | 1.33 | 2.025 | 3.66 | 2.04 | 2.52 | 1.91 | 2.28 | 2.28 |
| **Cl/F (mL/min/kg)** | 45.31 | 22.16 | 33.74 | 61.01 | 34.04 | 42.03 | 31.84 | 38.04 | 38.04 |

Abbreviations: $k_{el}$ = terminal rate constant; $t_{1/2}$ = half-life [Data points used to calculate $k_{el}$ are bolded in S2 Table in S1 File]; $T_{max}$ time to reach maximal plasma concentration; $C_{max}$ = maximal plasma concentration; $AUC_{0-12\ h}$ = area under the plasma concentration time curve form 0 hr to 12 hr after tramadol dosing; $AUMC_{0-12\ h}$ = area under the moment curve for 12 h after tramadol dosing; MRT = mean residence time; Vz/F apparent volume of distribution, Cl/F = apparent clearance;

* only determined for indices that are not dose dependent.

[a] p value when tramadol 2 mg/kg dose vs 4 mg/dose values for this indice are compared by one-way ANOVA.

**Table 2. Pharmacokinetic indices of M1 when tramadol was administered at 2 mg/kg, and 4 mg/kg both administered by s.c. injection.**

| Indice | Tramadol administered @ 2 mg/kg | | | Tramadol administered @ 4 mg/kg | | | | |
|---|---|---|---|---|---|---|---|---|
| | Koala 1 (K1) (M) | K2 F | Median | K3 (M) | K4 (M) | K 5 (F) | K 6 (F) | Median |
| $k_{el}$ (1/h) | 0.07 | 0.04 | 0.06 | 0.11 | 0.02 | 0.03 | 0.03 | 0.03 |
| $t_{1/2}$ (h) | 10.65 | 17.48 | 14.07 | 6.54 | 44.43 | 23.74 | 25.64 | 24.69 |
| $T_{max}$ (h) | 4.0 | 6.0 | 5.0 | 2.0 | 2.0 | 8.0 | 6.0 | 4.0 |
| $C_{max}$ (ng/mL) | 34.28 | 40.34 | 37.31 | 78.49 | 213.02 | 83.49 | 95.15 | 88.82 |
| $C_{max}$/ tramadol dose (ng/mL)/mg | 1.88 | 2.94 | 2.34 | 2.48 | 6.96 | 3.00 | 3.17 | 2.93 |
| $AUC_{0-12h}$ (ng/mL*h) | 314.64 | 375.62 | 345.13 | 663.13 | 1335.17 | 799.08 | 882.90 | 840.99 |
| $AUC_{0-12h}$ / tramadol dose (ng/mL*h)/mg | 17.29 | 27.42 | 21.64 | 20.99 | 43.63 | 28.74 | 29.43 | 27.76 |
| Median AUC M1: Median AUC tramadol | 345.13: 1049.81 = 0.33 | | | 840.99: 1669.29 = 0.50 | | | | |

Abbreviations: $k_{el}$ = terminal rate constant; $t_{1/2}$ = half-life life [Data points used to calculate $k_{el}$ are bolded in S3 Table in S1 File]; $T_{max}$ = time to reach maximal plasma concentration; $C_{max}$ = maximal plasma concentration; $AUC_{0-12\,h}$ = area under the plasma concentration time curve from 0 hr to 12 hr after tramadol dosing. Tramadol total dose for each koala is provided in Table 1. p values when tramadol 2 mg/kg dose vs 4 mg/dose when M1 values compared by one way ANOVA: $C_{max}$ /dose– p = 0.54; $AUC_{0-t}$/dose–p = 0.49.

**Table 3. Pharmacokinetic indices of M2 when tramadol was administered at 2 mg/kg, and 4 mg/kg, both administered by s.c. injection.**

| Indice | Tramadol administered @ 2 mg/kg | | | Tramadol administered @ 4 mg/kg | | | | |
|---|---|---|---|---|---|---|---|---|
| | Koala 1 (K1) (M) | K2 F | Median | K3 (M) | K4 (M) | K 5 (F) | K 6 (F) | Median |
| $k_{el}$ (1/h) | 0.08 | 0.09 | 0.09 | 0.16 | 0.08 | 0.09 | 0.10 | 0.10 |
| $t_{1/2}$ (h) | 8.42 | 8.00 | 8.21 | 4.37 | 8.23 | 7.93 | 6.93 | 7.43 |
| $T_{max}$ (h) | 4.00 | 4.00 | 4.00 | 1.00 | 2.00 | 1.00 | 4.00 | 1.50 |
| $C_{max}$ (ng/mL) | 50.36 | 75.39 | 62.88 | 99.47 | 162.35 | 126.05 | 143.47 | 132.76 |
| $C_{max}$/ tramadol dose (ng/mL)/mg | 2.77 | 5.50 | 3.94 | 3.15 | 5.31 | 4.53 | 4.78 | 4.38 |
| $AUC_{0-12h}$ (ng/mL*h) | 470.93 | 687.51 | 579.22 | 651.00 | 1369.28 | 1041.88 | 1338.62 | 1190.25 |
| $AUC_{0-12h}$ / tramadol dose (ng/mL*h)/mg | 25.88 | 50.18 | 36.31 | 20.60 | 44.75 | 37.48 | 44.62 | 39.28 |

Abbreviations: $k_{el}$ = terminal rate constant; $t_{1/2}$ = half-life life [Data points used to calculate $k_{el}$ are bolded in S4 Table in S1 File]; $T_{max}$ = time to reach maximal plasma concentration; $C_{max}$ = maximal plasma concentration; $AUC_{0-12\,h}$ = area under the plasma concentration time curve from 0 hr to 12 hr after tramadol dosing. Tramadol total dose for each koala is provided in Table 1. p values when tramadol 2 mg/kg dose vs 4 mg/dose when M2 values compared by one way ANOVA: $C_{max}$ /dose– p = 0.81; $AUC_{0-t}$/dose–p = 0.92.

**Table 4. Comparison of tramadol parameters and M1 half-life (mean ± SD [range]) between species.**

| | Clearance mL/kg/min | Volume of distribution L/kg | Tramadol elimination half-life h | M1 half-life (h) |
|---|---|---|---|---|
| Koala s.c. | 39.40 ± 13.36[t] | 8.60 ± 2.33[t] | 2.65 ± 1.03[*] | 25.09 ± 0.04[*] |
| | [22.16–61.01] | [6.58–12.32] | [1.31–3.53] | |
| Cats [5] | 20.8 ± 3.2 [13.1–31.3] | 5.1 ± 0.3 [4.5–6.4][**] | [Approx. 1–3] | 4.5 |
| Dogs [10] | 54.63 ± 8.19 | 3.79 ± 0.93 | 1.80 ± 1.2 | 1.7 |
| Horses [34] | 26 ± 3 | 2.17 ± 3 | 1.37 ± 0.17 | |
| Humans [33] | 7 to 8 | 2.7[t] | 5.5 | 6–7 |

[t] apparent clearance or apparent volume of distribution;

[*] 4 mg/kg s.c. dosage;

[**]oral dosing

## Prediction of dose and analgesic efficacy

The minimum effective plasma concentration for tramadol in humans is somewhat variable depending on the study's experimental design. Therefore some minimal effective analgesic concentrations of tramadol in humans range from 100 ng/mL [24], with a median of 287.7 ng/mL [28], to as high as 590 ± 410 ng/mL [27]. The results suggest that 2 mg/kg s.c. administered to koalas does not convincingly meet the effective human analgesic M1 plasma concentration (as demonstrated in Fig 4) but does, when administered at 4 mg/kg s.c. ($C_{max} > 94$ ng/mL). Another marker of analgesic efficacy is reported the M1: tramadol AUC ratio [5, 10, 12]. This ratio in the cat is 1–1.2 [5], 0.25 in humans [12] and 0.003 in the dog [10]. The ratio calculated in this study, in the koala, was 0.50 when administered at 4 mg/kg s.c. and 0.33 when administered at 2 mg/kg s.c.

This study was a pharmacokinetic study and could not directly determine whether the 2 mg/kg or 4 mg/kg had an analgesic effect in the koalas recruited. The effective analgesia plasma concentration was extrapolated from that for humans to compare with the plasma concentrations in the koala. Studies supporting tramadol's analgesic activity in non-human species are limited and are generally equivocal [37–39]. It is possible that the tramadol / M1 effective plasma analgesic concentrations may differ across species. Tramadol has been trialed in koalas (5mg/kg via intravenous administration) with moderate to severe pain (cases of long bone fractures and soft tissue injury) without apparent analgesic effect [40]. However, the 4 mg/kg s.c. dosage administered twice daily, was used to treat many of the surviving koalas burnt in the Australian bush-fire season 2019 / 2020, and anecdotal reports suggest effective analgesia, as koalas were less likely to exhibit signs of pain (teeth grinding, ear flicking, tachypnoea, grunting, crying and sweaty palms [13]). One of the authors, uses this dose routinely to control pain in some injured Australian wildlife with apparent efficacy.

The analgesic efficacy of tramadol is conventionally determined to be attributable to the amount of M1 formed by the biotransformation of tramadol. However, tramadol is also recognised to provide some analgesia by inhibiting the re-uptake of noradrenaline and serotonin into the presynaptic neurons and activating the descending pain inhibition pathway [38]. The importance of stimulating this pathway to provide some analgesia in the koala is not known.

In humans it is generally considered that tramadol undergoes biotransformation to five metabolites [8, 9], however over 20 metabolites have been identified in dogs [38]. Numerous metabolite peaks were seen after tramadol administration in koalas' chromatograms (Fig 1C and 1D). M1 and M2 were formally identified in the chromatograms from the treated koalas, as analytical standards for these metabolites are available. However, it is possible that one or more of the other metabolites may provide some analgesic activity in the koala. Another issue in those species that produce multiple metabolites is the possible co-elution of one or more metabolites such as M1 on the chromatogram, artificially inflating the peak area which is used for drug quantification [10]. The assay used here was developed to ensure that the chromatographic peaks for tramadol, M1 and M2 were well separated from each other and any other likely metabolite or endogenous plasma peak. When peaks overlapped, such as those located at M1's retention time in chromatograms for two of the male koalas, LC-MS was used to identify the components of both peaks.

Koalas administered with tramadol at 4 mg/kg did not demonstrate any obvious clinical changes in their behavior other than initial mild sedation after drug administration. Although tramadol is considered to have minimal effects on the gastrointestinal tract, long-term administration may cause constipation or diarrhoea infrequently in some species such as in the dog [41]. An aged cat was erroneously dosed with tramadol at 80 mg/kg (the intended dose was 4 mg/kg) with the cat displaying signs suggestive of serotonin syndrome [42]. The median lethal

dose ($LD_{50}$) when tramadol is administered orally to the rat is 300 mg/kg [43]. Toxic dosages can result in biochemical and histological abnormalities in the liver, kidney, brain, heart and lung [43]. Due to interspecies differences, the $LD_{50}$ for the koala cannot be predicted. Additionally, this study's results demonstrate that M1 concentrations may increase substantially with increasing tramadol dosage and therefore it is possible that multiple or higher tramadol doses may significantly increase M1 concentrations, also potentially resulting in adverse / toxic effects.

Tramadol is available in oral and injectable formulations for administration to humans. There is substantial evidence that oral medications (such as enrofloxacin and meloxicam) are poorly absorbed in koalas [1, 17, 18, 20]. Tramadol was administered s.c. as this route appears to have superior bioavailability than oral administration. However, a future study on oral administration of tramadol may be warranted as it is possible that, due to the first-pass metabolism effect, orally absorbed tramadol could rapidly result in higher M1 concentrations.

The results of this study support the hypothesis that injectable tramadol at 4 mg/kg may provide analgesia for the koala. Unlike meloxicam s.c. administration, M1 seems to have a useful half-life if medicated tramadol is administered at the 4 mg/kg dosage. Tramadol is a complex molecule and in some parts of the world e.g. United Kingdom, M1 (desmetramadol) is now available as an injectable formulation. Such a formulation may prove to be a useful analgesic for the koala in the future.

## Supporting information

**S1 File.**
(XLSX)

## Acknowledgments

The authors thank the participation of Taronga Zoo veterinarians, laboratory technical officers, veterinary nurses, and koala keepers. Thanks to Paul Thompson at Taronga Zoo for the preparation and shipping of the samples. Thanks to Dr Ray Austen, Keen St Veterinary Clinic for providing additional plasma samples for this study.

## Author Contributions

**Conceptualization:** Larry Vogelnest, Merran Govendir.

**Data curation:** Larry Vogelnest, Peter Valtchev, Merran Govendir.

**Formal analysis:** Benjamin Kimble, Peter Valtchev, Merran Govendir.

**Funding acquisition:** Merran Govendir.

**Investigation:** Benjamin Kimble, Larry Vogelnest, Peter Valtchev, Merran Govendir.

**Methodology:** Benjamin Kimble, Larry Vogelnest, Peter Valtchev.

**Project administration:** Larry Vogelnest, Merran Govendir.

**Resources:** Peter Valtchev, Merran Govendir.

**Validation:** Benjamin Kimble.

**Writing – original draft:** Benjamin Kimble.

**Writing – review & editing:** Benjamin Kimble, Larry Vogelnest, Peter Valtchev, Merran Govendir.

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
