## [Decision Letter · Decision Letter 0]

13 Jan 2021

PONE-D-20-36088

Pharmacokinetic profile of injectable tramadol in the koala (Phascolarctos cinereus) and prediction of its analgesic efficacy

PLOS ONE

Dear Dr. Govendir,

Thank you for submitting your manuscript to PLOS ONE. After careful consideration, we feel that it has merit but does not fully meet PLOS ONE’s publication criteria as it currently stands. Therefore, we invite you to submit a revised version of the manuscript that addresses the points raised during the review process.

We look forward to receiving your revised manuscript.

Kind regards,

Thomas P.C. Dorlo, PhD

Academic Editor

PLOS ONE

Journal Requirements:

2.) Please include a copy of Table 5 which you refer to in your text on page 13.

3.) Please include captions for your Supporting Information files at the end of your manuscript, and update any in-text citations to match accordingly. Please see our Supporting Information guidelines for more information: http://journals.plos.org/plosone/s/supporting-information.

4.) We note you have included a table to which you do not refer in the text of your manuscript. Please ensure that you refer to Table 4 in your text; if accepted, production will need this reference to link the reader to the Table.

5.) We noticed you have some minor occurrence of overlapping text with the following previous publication(s), which needs to be addressed:

- https://onlinelibrary.wiley.com/doi/abs/10.1111/jvp.12767

- https://www.tandfonline.com/doi/abs/10.1080/00498254.2019.1697014?journalCode=ixen20

- https://ses.library.usyd.edu.au/handle/2123/18012

In your revision ensure you cite all your sources (including your own works), and quote or rephrase any duplicated text outside the methods section. Further consideration is dependent on these concerns being addressed.

Reviewers' comments:

Reviewer's Responses to Questions

**Comments to the Author**

1. Is the manuscript technically sound, and do the data support the conclusions?

Reviewer #1: Yes

Reviewer #2: Yes

Reviewer #3: Yes

2. Has the statistical analysis been performed appropriately and rigorously? 

Reviewer #1: Yes

Reviewer #2: Yes

Reviewer #3: Yes

3. Have the authors made all data underlying the findings in their manuscript fully available?

Reviewer #1: Yes

Reviewer #2: Yes

Reviewer #3: Yes

4. Is the manuscript presented in an intelligible fashion and written in standard English?

Reviewer #1: Yes

Reviewer #2: Yes

Reviewer #3: Yes

5. Review Comments to the Author

Reviewer #1: I have only minor comments

Table 1 and 2: I am not sure reporting SD when there are only two or four observations is meaningful.

“the amount of each enantiomer may have a different rate of elimination” seem to suggest non-linear kinetics which I do not think is the intention here. I suggest removing “the amount of”.

Page 13 line 265, the units are wrong? The text mentions a half-life while the number mentions are mg/ml.

Reviewer #2: This is a well written manuscript describing a fairly simple pharmacokinetic study of tramadol in koala's. This type of research is essential to ensure the safe and effective use of analgesics in different animal species. I have a few minor comments and questions:

1. In Tables 1 through 3 it would be helpful if AUC and Cmax were reported as AUC/Dose and Cmax/Dose as well so that readers can evaluate whether there is dose-linearity.

2. I could not evaluate Figures 2 and 3 because they were black in my manuscript.

3. You argue that subcutaneous administration is preferable in koala's due to better absorption. However, if analgesia is primarily due to the M1 metabolite, wouldn't oral administration potentially be preferable due to the first-pass effect giving higher and more rapid M1 concentrations? Shouldn't this be included in the discussion?

Reviewer #3: Comments to the authors:

Koalas are facing many threats and new knowledge about safe and effective analgesia that can be used in koalas is very important. The authors have conducted a well described pharmacokinetic study of tramadol in koalas that is expected to add useful information to the literature.

Minor comments

1. The authors are comparing the minimum effective concentrations of tramadol and M1 in koalas to those reported in humans. Also, protein binding was measured in koala plasma. In the manuscript it is not always clear whether the total or unbound concentrations were compared (line 313 mentioned unbound plasma concentrations). Since plasma protein binding of tramadol M1 appears to be similar between koalas and humans, comparing total concentrations seems fine, however this could be briefly stated. In addition to the comparison with effective concentrations, a comparison with potentially toxic concentrations could be useful, although there would be limitations because of inter species differences.

2. The apparently longer elimination half-life of M1 in koalas compared to humans is interesting however does not appear to be due to protein binding, based on the information given in the manuscript, because plasma protein binding of tramadol M1 appears to be similar between koalas and humans.

3. The profiles in Figure 4 indicate that the M1 concentrations may increase more than proportionally with tramadol dose, or this could be due to interindividual variability between koalas, however it would be useful to discuss. If M1 concentrations increase more than dose proportionally it could potentially have an impact for concentrations reached with multiple dosing or higher doses, or potentially impact toxicity in case doses are increased.

4. The average profiles in Figure 3 show the concentrations following the higher dose dropping slightly below those of the lower dose, most likely due to the interindividual variability between koalas. This might be worth stating in the figure legend or footnote. Also, it should be reported whether the concentrations of 12 h for the higher dose were all below the limit of quantification and are therefore not in Figure 3.

5. Line 168: elimination constant should be elimination rate constant, also k is the inverse slope of the elimination or terminal part of the semi-log curve.

6. It would be useful to report how many time points were used to determine k and which software was used for noncompartmental analysis.

7. The M1 has a quite flat profile between 2 and 12 h and it might be useful to show the profiles on log scale e.g. in the supplementary information to illustrate the half-life. There is likely a limitation in the determination of the elimination half-life of M1, because the average of 25 h reported for the 4 mg/kg dose is twice the duration of the study (last sample at 12 h), so this might be mentioned as a limitation in the discussion.

8. Line 254: the words “the amount of” seem not to be needed.

9. Lines 316 to 320: The sentence should be reworded to improve clarity.

10. Line 323: the word “range” seems not to be needed.

11. It is not clear whether all sheets (e.g. sheet 2) of the supplementary excel file are intended to be included in the supplementary material for readers.

6. PLOS authors have the option to publish the peer review history of their article (what does this mean?). If published, this will include your full peer review and any attached files.

Reviewer #1: No

Reviewer #2: **Yes: **Ronette Gehring

Reviewer #3: No

---

## [Author Response · Author response to Decision Letter 0]

8 Feb 2021

The Authors thank the Editor and all three Reviewers for their time and their excellent comments and suggestions. 

Editor’s comments 

• The manuscript has been edited to meet the PLOS ONE’s style requirements 

2.) Please include a copy of Table 5 which you refer to in your text on page 13.

• Apologies, there was no Table 5; the reference was meant to be Table 4, now corrected. See lines 315 and 322 

3.) Please include captions for your Supporting Information files at the end of your manuscript, and update any in-text citations to match accordingly. Please see our Supporting Information guidelines for more information: http://journals.plos.org/plosone/s/supporting-information.

• This has been done to meet PLOS ONE’s supporting information guidelines

4.) We note you have included a table to which you do not refer in the text of your manuscript. Please ensure that you refer to Table 4 in your text; if accepted, production will need this reference to link the reader to the Table.

• This has been corrected according to the Editor’s comment 2. Additionally, a further sentence has been added to introduce Table 4. See line 321: A summary of tramadol parameters available for some species are provided in Table 4. 

5.) We noticed you have some minor occurrence of overlapping text with the following previous publication(s), which needs to be addressed:

- https://onlinelibrary.wiley.com/doi/abs/10.1111/jvp.12767

- https://www.tandfonline.com/doi/abs/10.1080/00498254.2019.1697014?journalCode=ixen20

- https://ses.library.usyd.edu.au/handle/2123/18012

In your revision ensure you cite all your sources (including your own works), and quote or rephrase any duplicated text outside the methods section. Further consideration is dependent on these concerns being addressed.

• This manuscript has been run through ‘Grammerly’ to check for overlapping text and has been edited to minimise. 

……………………………………………………………………………………………………………………………………………………..

Reviewer #1: I have only minor comments

The Authors thank Reviewer 1 for their helpful comments. 

Table 1 and 2: I am not sure reporting SD when there are only two or four observations is meaningful.

The mean ± SD has been removed and replaced by a median column for the 4 mg/kg tramadol administration. The reader can see the range as all the values from all koalas are presented. 

• Values in the text and in the Tables have been changed to medians in the text 

“the amount of each enantiomer may have a different rate of elimination” seem to suggest non-linear kinetics which I do not think is the intention here. I suggest removing “the amount of”.

• Thank you for this comment. “The amount of” has been deleted 

Page 13 line 265, the units are wrong? The text mentions a half-life while the number mentions are mg/ml.

Thank you for detecting this. Accordingly, the median half-lives units for M1 and tramadol have been changed to hours (h) - see lines 297 and 298

…………………………………………………………………………………………………………………………………………………………

Reviewer #2: 

The Authors thank Reviewer 2 for their helpful comments. 

This is a well written manuscript describing a fairly simple pharmacokinetic study of tramadol in koala's. This type of research is essential to ensure the safe and effective use of analgesics in different animal species. I have a few minor comments and questions:

1. In Tables 1 through 3 it would be helpful if AUC and Cmax were reported as AUC/Dose and Cmax/Dose as well so that readers can evaluate whether there is dose-linearity.

• As requested, AUC / Dose and Cmax/ dose have been added to Tables 1 through 3. A one-way ANOVA was undertaken to analyse if there was a significant difference for Cmax/dose and AUC 0-t / dose for tramadol, M1 and M2. There was no significant difference for any comparison. Non-significant p values are provided with in Table 1 (tramadol) and in footnotes for M1 and M2 Tables. 

The following has been added to the Material and Methods see lines 201 – 204 

In order to assess any difference in Cmax or AUC0-t h when normalised for dose (2 mg/kg or 4 mg/kg tramadol administration); the Cmax / dose and AUC0-t / dose values for tramadol, M1 and M2 were compared by a one-way analysis of variance (ANOVA) using Graphpad Prism 9.0 (San Diego, CA). The level of significance (p) was < 0.05. 

The following has been added to the Results see lines 275 - 276: 

There were no significant differences in the tramadol, M1 or M2 Cmax or AUC0-t when normalised for tramadol dose (2 mg/kg or 4 mg/kg administration). 

The following has been added to the Discussion see lines 301 - 303: 

There were no statistical differences when the Cmax or AUC0-t for tramadol, M1 or M2 were normalised for tramadol dose, suggesting that tramadol, M1 and M2 demonstrate linear pharmacokinetics regardless to tramadol dose.

2. I could not evaluate Figures 2 and 3 because they were black in my manuscript.

• Apologies, for this this problem. CLICK ON THE LINK IN THE TOP RIGHT OF EACH FIGURE AND IT WILL OPEN IN A NEW WINDOW AND EACH FIGURE / GRAPH IS THEN VISIBLE. 

3. You argue that subcutaneous administration is preferable in koala's due to better absorption. However, if analgesia is primarily due to the M1 metabolite, wouldn't oral administration potentially be preferable due to the first-pass effect giving higher and more rapid M1 concentrations? Shouldn't this be included in the discussion?

• Thank you for this suggestion. Accordingly, the following has been added, see lines 409-411 : 

However, a future study on oral administration of tramadol may be warranted as it is possible that, due to the first-pass metabolism effect, orally absorbed tramadol could rapidly result in higher M1 concentrations. 

………………………………………………………………………………………………………………………………………………

Reviewer #3: 

The Authors thank Reviewer 3 for their helpful comments. 

Comments to the authors:

Koalas are facing many threats and new knowledge about safe and effective analgesia that can be used in koalas is very important. The authors have conducted a well described pharmacokinetic study of tramadol in koalas that is expected to add useful information to the literature.

Minor comments

1. The authors are comparing the minimum effective concentrations of tramadol and M1 in koalas to those reported in humans. Also, protein binding was measured in koala plasma. In the manuscript it is not always clear whether the total or unbound concentrations were compared (line 313 mentioned unbound plasma concentrations). Since plasma protein binding of tramadol M1 appears to be similar between koalas and humans, comparing total concentrations seems fine, however this could be briefly stated. In addition to the comparison with effective concentrations, a comparison with potentially toxic concentrations could be useful, although there would be limitations because of inter species differences.

Re plasma protein binding in line 313, it was the total binding that was being compared. However, in the light of your comment, incorporating plasma protein binding at this point of the manuscript may confuse the reader and thus has now been deleted.

*Re the potentially toxic concentrations the following statement has been added: (see lines 396 – 401) 

An aged cat was erroneously dosed with tramadol at 80 mg/kg (the intended dose was 4 mg/kg) with the cat displaying signs suggestive of serotonin syndrome [42]. The median lethal dose (LD50) when tramadol is administered orally to the rat is 300 mg/kg [43]. Toxic dosages can result in biochemical and histological abnormalities in the liver, kidney, brain, heart and lung [43]. Due to interspecies differences, the LD50 for the koala cannot be predicted.

2. The apparently longer elimination half-life of M1 in koalas compared to humans is interesting however does not appear to be due to protein binding, based on the information given in the manuscript, because plasma protein binding of tramadol M1 appears to be similar between koalas and humans.

This is a helpful comment. The plasma protein binding was undertaken to attempt to see if this could partially explain the longer M1 half-life in the koala. It is agreed that the tramadol and M1 binding in the koala and human are similar. Therefore, the phrase “it is interesting to speculate whether the higher M1 binding to plasma proteins observed here (75%) contributed to the long M1 half-life in the koala has been deleted”. 

3. The profiles in Figure 4 indicate that the M1 concentrations may increase more than proportionally with tramadol dose, or this could be due to interindividual variability between koalas, however it would be useful to discuss. If M1 concentrations increase more than dose proportionally it could potentially have an impact for concentrations reached with multiple dosing or higher doses, or potentially impact toxicity in case doses are increased.

Considering your comment this has been added, see lines 401 to 404: 

Additionally, the M1 results in this study demonstrate that M1 concentrations may increase substantially with increasing tramadol dosage and therefore it is possible that multiple or higher tramadol doses may significantly increase M1 concentrations, also potentially resulting in adverse / toxic effects. 

4. The average profiles in Figure 3 show the concentrations following the higher dose dropping slightly below those of the lower dose, most likely due to the interindividual variability between koalas. This might be worth stating in the figure legend or footnote. Also, it should be reported whether the concentrations of 12 h for the higher dose were all below the limit of quantification and are therefore not in Figure 3.

The legend for Figure 3 has been emended accordingly: The graph of the 4 mg/kg tramadol dose dips slightly below the 2 mg/kg tramadol dose – this may be due to individual variability between koalas. The tramadol concentrations of the 4 mg/kg tramadol dose at not provided at 12 h as they are below the LLOQ = 15.63 ng/mL. 

5. Line 168: elimination constant should be elimination rate constant, also k is the inverse slope of the elimination or terminal part of the semi-log curve.

This has been emended accordingly. 

6. It would be useful to report how many time points were used to determine k and which software was used for noncompartmental analysis.

PK Solver [22] was used for the noncompartmental analysis see lines 179 – 180.

The number of time points used to determine K are now bolded in S2 to S4 tables. The reader is directed there by the information under Tables 2, 3 and 4. E.g. See Table 2: Abbreviations: kel = terminal rate constant; t1/2 = half-life [Data points used are bolded in S2 Table];

7. The M1 has a quite flat profile between 2 and 12 h and it might be useful to show the profiles on log scale e.g. in the supplementary information to illustrate the half-life. There is likely a limitation in the determination of the elimination half-life of M1, because the average of 25 h reported for the 4 mg/kg dose is twice the duration of the study (last sample at 12 h), so this might be mentioned as a limitation in the discussion.

The M1 figure (Figure 4) has been modified to show the log concentration. 

The following has been added to the Discussion, see lines 292 – 296:

One of the limitations of interpretation of the M1 data was that the last data point collected was at 12 h, indicating that at 12 h, M1 was in the early elimination phase in all koalas. Further plasma collection at time points beyond 12 h of tramadol administration would provide a more accurate determination of the M1 plasma profile.

8. Line 254: the words “the amount of” seem not to be needed.

Reviewer 1 also commended on this and “the amount of” has been deleted. 

9. Lines 316 to 320: The sentence should be reworded to improve clarity.

This sentence has been reworded as (now see Lines 346 – 349): 

The minimum effective plasma concentration for tramadol in humans is somewhat variable depending on the study’s experimental design. Therefore some minimal effective analgesic concentrations of tramadol in humans range from 100 ng/mL [37], with a median of 287.7 ng/mL [27], to as high as 590 � 410 ng/mL [26].

10. Line 323: the word “range” seems not to be needed.

“range” has been deleted 

11. It is not clear whether all sheets (e.g. sheet 2) of the supplementary excel file are intended to be included in the supplementary material for readers.

Thank you. The extraneous information on sheet 2 of the supplementary excel file has been deleted. 

---

## [Editor Report · Decision Letter 1]

10 Feb 2021

Pharmacokinetic profile of injectable tramadol in the koala (Phascolarctos cinereus) and prediction of its analgesic efficacy

PONE-D-20-36088R1

Dear Dr. Govendir,

We’re pleased to inform you that your manuscript has been judged scientifically suitable for publication and will be formally accepted for publication once it meets all outstanding technical requirements.

Kind regards,

Thomas P.C. Dorlo, PhD

Academic Editor

PLOS ONE
---

## [Editor Report · Acceptance letter]

15 Feb 2021

PONE-D-20-36088R1 

Pharmacokinetic profile of injectable tramadol in the koala (*Phascolarctos cinereus*) and prediction of its analgesic efficacy   

Dear Dr. Govendir:

I'm pleased to inform you that your manuscript has been deemed suitable for publication in PLOS ONE. Congratulations! Your manuscript is now with our production department. 

Kind regards, 

on behalf of

Dr. Thomas P.C. Dorlo 

Academic Editor

PLOS ONE